# Research on Tribological Characteristics of Hard Metal WC-Co Tools with TiAlN and CrN PVD Coatings for Processing Solid Oak Wood

**Deividas Kazlauskas [1,\*], Vytenis Jankauskas [1,\*] and Simona Tučkutė [2]**

[1] Institute of Mechanical Engineering, Vytautas Magnus University, K. Donelaičio st. 58, Kaunas 44248, Lithuania

[2] Center for Hydrogen Energy Technologies, Lithuanian Energy Institute, Breslaujos st. 3, Kaunas 44403, Lithuania; simona.tuckute@lei.lt

**\*** Correspondence: deividaskazlauskas78@gmail.com (D.K.); vytenisjankauskas@vdu.lt (V.J.)

**Abstract:** The article presents research results that demonstrate the impacts of mechanical and tribological characteristics on the efficiency of cutting tools made from hard metal WC-Co. Uncoated tools and tools coated with physical vapor deposition (PVD) coatings (CrN and TiAlN) were tested. The thickness of coatings was determined, and tests of roughness, microhardness, and adhesion were performed. The coefficient of friction (COF) was established by different methods, and the wear of the tested cutters which occurs during the milling of solid oak wood, and the impact of this on surface roughness, were determined. The results revealed that uncoated WC-Co cutters are the least resistant to wear, while cutters coated with chromium nitride (CrN) are the most resistant. Both PVD coatings were damaged in the 9050 m in area of the cutting edge.

**Keywords:** hard metal tool; PVD coatings; wear of tools; wood milling; oak wood

## 1. Introduction

The turnover of the wood industry in the European Union in 2010–2011 exceeded 122 billion euros [1], and the amount of wood processed in the EU grew from $3.8 \times 10^8$ to $4.9 \times 10^8$ m$^3$ from 2009 to 2018 [2]. A significant factor in the value of wood products is the cost of raw materials, energy, work, tools, and machinery.

The cutting of wood and its materials is a complex process [3]. This is due to the specific characteristics of wood, including anisotropy, internal stresses, the presence of solid mineral particles, and the variable cross-sectional density [3]. The processing of wood and its composite materials is characterized by several factors: high processing speeds (20–40 m·s$^{-1}$), the need for especially sharp edges on cutting tools and sharp cutting angles in order to achieve better quality, and the low thermal conductivity of wood [3,4]. Cutting tools are subjected to large mechanical and thermal loads, plastic deformation, adhesive and diffusive wear, and oxidation [5–9]. Therefore, their efficiency depends on their resistance to mechanical loads, the stability of their physical and chemical characteristics, their structure, and the characteristics of their superficial layers [10,11].

High speed steel and alloy-treated steel tools get worn out quickly; therefore, their use results in reduced output, increased energy expenses, and a decrease in the quality of the processed surface. In order to reduce the wear of tools, hard metal is used for wood processing. Tough, hard metals containing 5%–15% cobalt are used to process massive pieces of wood, while harder metals that contain up to 5% cobalt and finer grains are used to process wood products (oriented strand boards, medium density fiber boards, wood–plastic composites, etc.) [12]. In order to increase the output of hard-metal tools, they are modified with hard coatings of micrometric thicknesses using physical vapor deposition (PVD) and chemical vapor deposition (CVD). The coatings protect the substrata of

the metal from mechanical wear and corrosion, while spreading heat and reducing stresses [13,14]. The requirements set for tool coatings are good adhesion with the substratum, high hardness at the relevant operating temperature, high strength, chemical inactivity of the processed material, a fine-grained crystalline microstructure, high residual stresses, a smooth surface morphology (without cracks), and a low thermal conductivity coefficient [15].

The PVD coatings increase the strength of the tool surfaces, improve their tribological and anticorrosive characteristics, and increase resistance to wear and the operating temperature [16–21]. It is probable that tools without PVD coatings will not be used for commercial purposes in 20 years [22–26].

PVD methods are suitable for coating wood processing tools because they qualitatively coat the cutting edges, where the radius is below 20 μm. Compared to CVD coatings, PVD coatings are more resistant to wear because they have hardness and compression stresses that provide additional hardness and reliability to the tool. When fine-grained (up to 1 μm) hard metal is used, the blades are even more resistant to wear [14], while the use of a PVD coating increases the strength even more [18,26,27]. The main coatings applied on tools for the treatment of wood and wood products are Ti, Cr, Al, and diamond like carbon (DLC) [10,13,28,29].

WC-Co tools coated with PVD are used in the wood industry when strong cutting tools and sharp edges are needed [30–32].

The main factors that affect the quality of coatings are hardness, adhesion, and thickness. To achieve coatings with high hardness values, nitrides, carbides, borides (<25 GPa), cubic boron nitride (approximately 48 GPa), diamond (70–100 GPa), DLC, and $CN_x$ are used [15,27–29,33]. Adhesion with the substratum is safeguarded by surface cleaning, using an appropriate roughness, and increasing the adhesion in the interlayer; this is assessed by a scratch test [34–36]. The coating thickness affects a tool's resistance to wear, output, and cutting forces. The optimal thickness is from 1 to 10 μm [15].

Various wood types are processed around the world. They are characterized by complex structures and characteristics. It is known that different wear mechanisms affect the cutting tools used to process different types of wood [36–39], so in order to reduce the costs of the tools and to increase the processing productivity, it is necessary to purposefully select tool coatings for specific types of wood so that minimal wear can be guaranteed. PVD coatings comprising Cr and Ti have been widely analyzed; however, studies on the resistance of hard-metal tools coated with these materials and used for wood milling are only now beginning to emerge, and have not been performed concerning the milling of oak wood.

The purpose of this research is to analyze the influence of the mechanical and tribological properties of CrN and TiAlN coatings formed using PVD methods on the wear of tools during oak wood processing. We conducted tests on the coating composition, structure, thickness, roughness, microhardness, and coefficient of friction, as well as carrying out scratch and industrial tests.

## 2. Materials and Methods

In terms of samples, we used WC-Co cutters made of hard metal T06MG (Table 1), measuring 30 mm × 12 mm × 1.5 mm and weighing 6.75 ± 0.03 g, from the manufacturer TIGRA GmbH (Gewerbering, Germany, Figure 1). The surface grinding trace with the cutting edge formed an angle δ of about 10°. These samples were coated with a 2-μm-thick PVD layer (CrN and TiAlN) by Cemecon Scandinavia A/S (Hinnerup, Denmark). The characteristics of the PVD coatings are presented in Table 2.

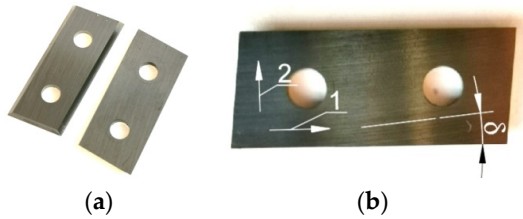

(**a**)                  (**b**)

**Figure 1.** Specimens: (**a**) Hard metal WC-Co (Co 6%) cutters; (**b**) Angle between the polishing trace and the cutting edge δ; 1: direction parallel to the cutting edge; 2: direction perpendicular to the cutting edge (the slip direction of the chip on the rake face).

**Table 1.** Hard metal characteristics [40].

| Hard Metal Grade | Binder Co, % | Size of WC Grains, μm | Hardness, HV 10 | Bending Strength, N/mm$^2$ | Fracture Toughness, $K_1C$/MPa·m$^{-1/2}$ | Operating Temperature, °C | Coefficient of Friction | Surface Roughness $R_a$, μm |
|---|---|---|---|---|---|---|---|---|
| T06MG | 6 | 0.7–1.0 | 1800 | 2700 | 8.4 | 800–1000 | 0.4–0.6 | 0.04 |

**Table 2.** Characteristics of physical vapor deposition (PVD) coatings [27,41–46].

| Coating | Number of Layers | Maximum Operating Temperature, °C | Hardness, GPa |
|---|---|---|---|
| TiAlN | Single layer | 800 | 30 ± 10 |
| CrN | Single layer | 700 | 18 |

Surface roughness and topography measurements were made using a stylus profilometer (MahrSurf GD 25, Mahr GmbH, Göttingen, Germany) with a stylus tip radius of 2 μm roughness and a measurement length of 5.6 mm. The mean of five measurements was recorded.

Microhardness was measured using a Micro Combi Tester (Anton Paar Switzerland AG, Aargau, Switzerland) device at a load of 50 mN (30 s load addition, 10 s maintenance, and 30 s removal time), and the mean of measurements 4–6 was recorded.

Scratch testing of coatings was performed by a sharp diamond Rockwell indenter on a CETR/Bruker UMT-2 tribometer (Bruker Corporation, Campbell, CA, USA) at a speed of 1 mm·min$^{-1}$. The load increased to 137.3 N. The conic angle of the indenter was 120°, and its radius was 200 μm. The scratch length was 7 mm. The test was performed by drawing lines parallel to the cutting edge. The results are shown as the mean of four measurements.

The static COF of oak wood and specimens was established using the inclined plane when the angle-changing speed was 5 min$^{-1}$, and the mean of 5 measurements was recorded.

The cutting edge radius ϱ was measured using an optical metallographic microscope (Eclipse MA-100, Nikon, Tokyo, Japan). The results are shown as the mean of five measurements.

The wear of the cutter's rake face *Wd* and damage caused to the coatings were analyzed using an optical metallographic microscope (Eclipse MA-100, Nikon), according to the scheme provided in Figure 2a. The wear *Wd* and roughness of the milled surface ($R_z$, μm) after the tool had worked cutting paths of 1, 1050, 2050, 3050, 6050, and 9050 m (up to 2.16 million cutting cycles of 4.2 mm each) were measured optically. The results are shown as the mean of five measurements. The cutting path is expressed as the basic cutting motion.

The roughness of the processed surface $R_z$ was measured along the wood fiber using a MahrSurf GD 25 contact profilometer (Mahr GmbH, Göttingen, Germany) (run 5 mm, speed 5 mm·s$^{-1}$), and the mean of five measurements was recorded.

PVD coating analysis was performed using a Hitachi S-3400N-II (Hitachi, Ltd., Tokyo, Japan) scanning electron microscope (SEM).

The studies on the wear of hard metal WC-Co cutters with and without TiAlN and CrN PVD coatings were performed while working with a fixed milling tool with one cutting tooth. The characteristics of the milling tool and the cutting tooth are provided in Table 3 and Figure 2. The scheme used for specimen milling is illustrated in Figure 2c.

**Table 3.** Characteristics of the fixed milling tool.

| Cutting Diameter $D$, mm | Length of the Fixed Milling Tool $l_2$, mm | Rake Angle $\gamma$, ° | Sharpness Angle $\beta$, ° | Clearance Angle $\alpha$, ° | Cutting Angle $\lambda$, ° |
|---|---|---|---|---|---|
| Ø18 | 55 | 27 | 53 | 10 | 87 |

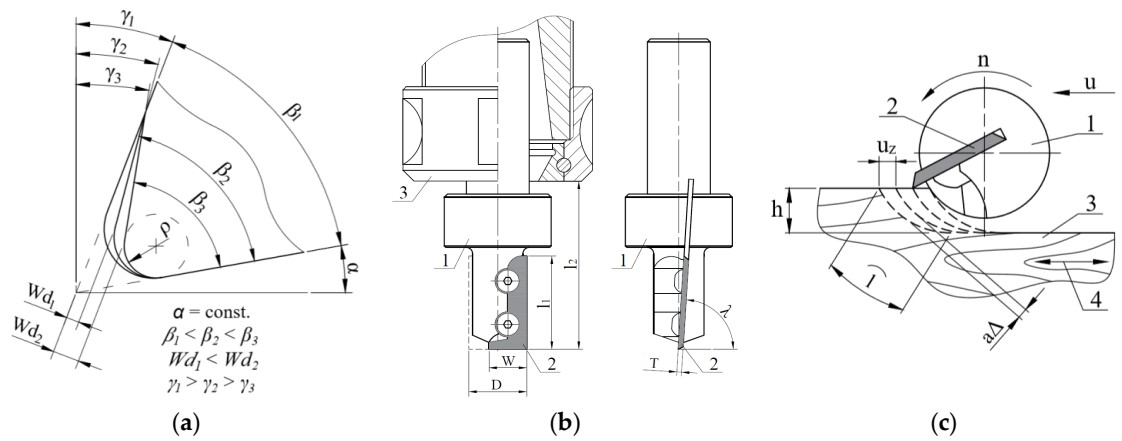

(**a**) (**b**) (**c**)

**Figure 2.** Cutting tool and down milling processing modes: (**a**) Measurement scheme of the wear of the cutting edge, $\alpha$: clearance angle; $\beta$: sharpness angle; $\gamma$: rake angle; $\varrho$: cutting edge radius; $Wd$: wear; (**b**) Fixation of cutter; 1: fixed milling tool; 2: cutter; 3: HSK 63F gripper device; (**c**) Down milling scheme: 1: milling tool; 2: cutter; 3: oak wood; 4: direction of the wood fiber; $h$: radial depth of cut; $u_z$: feed per tooth; $u$: feed; $n$: rotation direction of the milling tool; $l$: length of contact arc; $a\Delta$: average shaving thickness.

The down milling process was chosen to obtain a better quality treated surface [47]. The wear of the milling tool cutters was examined while milling glued oak-wood panels (1000 mm × 1000 mm × 20 mm) that were made after gluing beams (1000 mm × 67 mm × 20 mm) using polyvinyl acetate dispersion (Danafix 437 D3, Dana Lim A/S, Køge, Denmark). The timber characteristics were as follows: average humidity $\omega = 8\%$, average number of rings in the trunk 1 cm–4.6 pcs., and density $\varrho$ = 737.8 kg/m³.

The Holzher ProMaster 7123 (Michael Weinig AG, Tauberbischofsheim, Germany) CNC machining center was used for milling at the modes presented in Table 4. The machining conditions were as follows: temperature in premises $t = 19 \pm 2$ °C, and relative air humidity $\psi = 60\% \pm 5\%$.

**Table 4.** Milling modes.

| Tool Rotation Speed $n$, $10^3$ min⁻¹ | Cutting Speed $v$, m·s⁻¹ | Thickness of Cut Layer $h$, mm | Feed per Cutter $u_z$, mm | Length of Contact Arch $l$, mm | Average Shaving Thickness $a\Delta$, mm | Width of Cut Layer $b$, mm |
|---|---|---|---|---|---|---|
| 18.0 | 17 | 1 | 0.1 | 4.2 | 0.024 | 6 |

## 3. Results

### 3.1. Choice of Coatings

CrN and TiAlN coatings were selected according to their characteristics on the basis on information from various sources [3,48,49].

The SEMs of TiAlN and CrN coatings are provided in Figure 3.

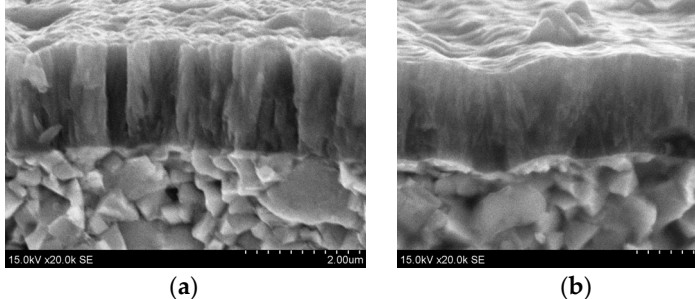

(**a**)  (**b**)

**Figure 3.** Breaks of hard metal WC-Co specimens with PVD coatings (×20,000): (**a**) CrN (1.83–1.89μm), (**b**) TiAlN (1.77–1.79 μm).

The average thickness of the PVD coatings, as determined by SEM, was 1.8–1.9 μm (Figure 3). Nitride coatings are characterized by a dense, fine-grained microstructure with low porosity, small internal stresses, excellent adhesion, chemical inertia, and a good thermal stability [50]. However, these coatings have a significant defect: their column structure. Micropores are present between the columns. The corrosive medium may reach a substratum through them and destroy it. The most effective method for improving the quality of the coatings is to form coatings where growth of the columns is prevented. This method reduces the possibility of corrosion appearing between the columns [51]. There is no literature about the influence of the direction of columns on the abrasive wear of the coating. However, Kondo et al. stated that dense column structures have better resistance to adhesive wear [52].

*3.2. Roughness Testing*

The roughness of the tool's rake face affects the roughness of the processed surface and the friction forces which occur during processing [53]. There is no doubt that the processing trace on the cutter surface and the cutter rake angle γ also affect the roughness of the processed surface.

When the surfaces are coated with a PVD coating, their roughness increases [54]. Uncoated hard metal WC-Co cutters have the least roughness. If measured in parallel to the cutting edge, $R_a$ = 0.04 μm, and if measured perpendicularly, $R_a$ = 0.08 μm (Figure 4). In terms of the roughness of specimens with PVD coatings, $R_a$ is bigger by 24.8%–31.4% along the cutting edge and by 3.7%–28.6% across the cutting edge, as shown in Figure 4.

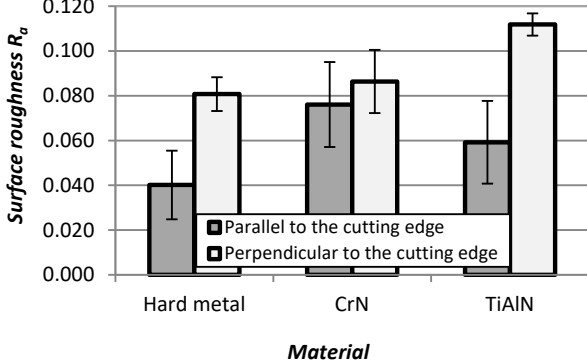

**Figure 4.** Roughness of WC-Co cutters with and without PVD coatings in the parallel and perpendicular directions to the cutting edge.

*3.3. Microhardness Testing*

Hard metal WC-Co (18 ± 1 GPa) and CrN coatings (19.4 ± 2 GPa) have similar levels of hardness. The hardness of TiAlN is greater (24.5 ± 5 GPa). The determined hardness of the hard metal WC-Co and CrN coating corresponds to the values reported in the literature (Table 2). However, the

hardness of the TiAlN coating was found to be lower than that reported in the literature (Table 2) due to its too deep indentation with respect to the coating thickness. This was due to the significantly softer substrate of the WC-Co [55]. The indentations of the hardness measuring indenter in the cutters of hard metal WC-Co with and without PVD coating are presented in Figure 5.

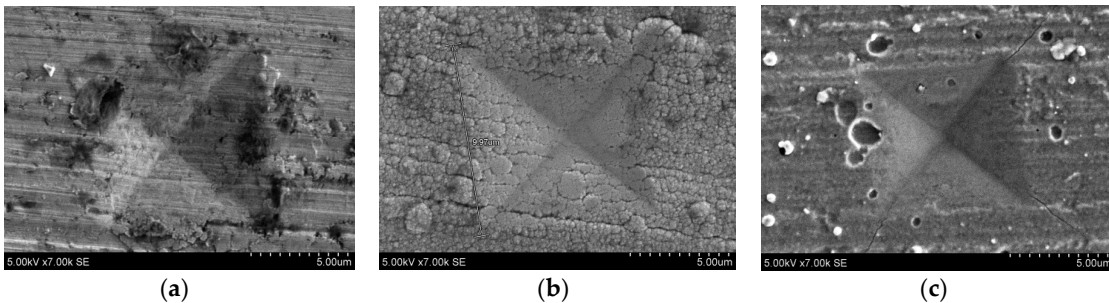

| (a) | (b) | (c) |

**Figure 5.** Trace of the Vickers indenter on surfaces (50 mN, ×7000): (**a**) Uncoated hard metal WC-Co; (**b**) With CrN coating; (**c**) With TiAlN coating.

The morphology of the uncoated WC-Co cutter surface is uneven. It has craters with an irregular form. The polishing trace and surface "tearing" defects can be seen clearly (Figure 5a). The indentation is clear.

CrN coating has a rough structure that is clearly different in the processing direction. Traces of the mechanical processing of the substratum are absent, and thus, the difference of the surface roughness parameter $R_a$ in the perpendicular direction is negligible (Figure 5b). The grooves of the surface processing trace are visible on the coating. Various forms of "grain" defects of 1–2 μm in size appear along the processing direction during the precipitation process, having a negative impact on the surface morphology, roughness, and coating durability [56]. The coating has a column structure (Figure 3a), so the corrosive medium may reach and destroy the substratum through the microgaps between the adjacent "columns" [57]. The hardness levels of CrN and WC-Co were similar, but the indentation left by the indenter on the CrN coating was more visible. There were cracks in the coating.

Pores of up to 1.5 μm in diameter and single "grain"-type defects (up to 1.0 μm) could be seen on the surface of the TiAlN coating (Figure 5c). Defects in the form of pores appeared along the entire thickness of the coating. They formed due to defects in the substratum (craters) that appeared mainly during polishing, or when micro- and macro- particles separate from the coating. These defects have a negative impact on the coating's resistance to corrosion and gas permeability [56,57]. The traces of mechanical processing of the substratum were indistinct. The indentations of the indenter were clear and had cracks in the corners. This shows the high level of hardness and fragility of the TiAlN coating.

*3.4. Scratch Test*

The scratches made in the hard metal WC-Co specimen with PVD coatings are presented in Figure 6.

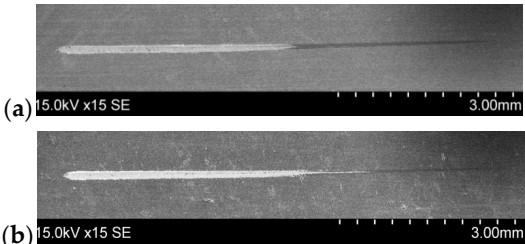

**Figure 6.** Scratch test results for PVD coatings (×15): (**a**) CrN coating; (**b**) TiAlN coating.

The characteristics of the coatings established by the scratch test are presented in Table 5.

**Table 5.** Characteristics of hard metal and PVD coatings established by scratch testing.

| Sample/Coating | COF When coating is damaged | Load for Appearance of Adhesion Damages $F_x$, N | Damage Width at Load of 137.3 N, μm |
|---|---|---|---|
| CrN | 0.14 | 44.7 | 118.8 |
| TiAlN | 0.16 | 32.8 | 121.9 |

CrN coating is more resistant to the scratching impact of the indenter, probably because of its plasticity. The load that generates damage during adhesion of the coating and the substratum was 44.7 N, while the width of the scratch on the coating was the smallest (118.8 μm). Meanwhile, the TiAlN coating was damaged by a much smaller indenter load: 32.8 N. The CrN coating had better adhesive parameters than TiAlN. This is very important for cutting tools [34–36].

The coefficients of friction of the surfaces of the diamond indenter and hard metal WC-Co with and without PVD coatings across the cutting edge are presented in Figure 7.

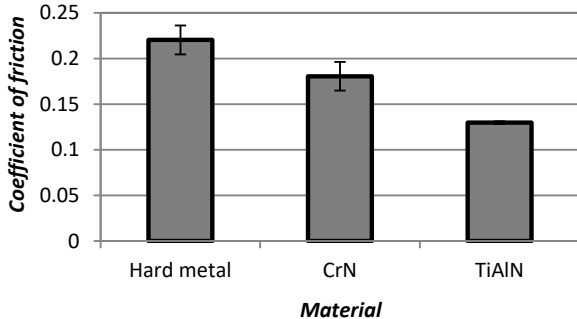

**Figure 7.** Coefficients of friction of the indenter and WC-Co specimens with and without PVD coatings.

The coefficients of friction of hard metal WC-Co and the indenter were the biggest (0.22), whereas those with CrN and TiAlN coatings were smaller by 18% and 28%, respectively. Therefore, when wood is processed using tools with PVD coatings, a smaller resistance to cutting is probable. This is also a precondition for reduced wear.

The static COFs of different cuts of oak wood (radial and tangential) and WC-Co with and without PVD coatings are presented in Figure 8.

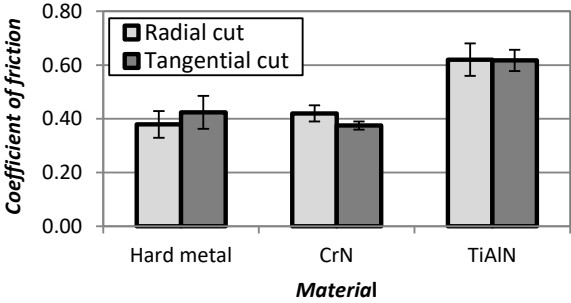

**Figure 8.** Static coefficient of friction between different cuts of oak wood (radial and tangential) and hard metal WC-Co with and without PVD coatings.

The static coefficients of friction did not differ significantly between oak wood and hard metal WC-Co with CrN coating. The cutting direction did not affect the coefficient of the friction of wood and hard metal WC-Co (with and without PVD coating) (Figure 8). The coefficient of friction of TiAlN coating and wood is the biggest (0.62 ± 0.03). Thus, it is probable that the greatest resistance to

cutting and wear exists in this case. The cutting direction has a negligible influence on the static coefficient of friction between the CrN coating and wood. For the radial section, it was 0.411, and for the tangent, it was 0.375 (Figure 8). Therefore, if oak wood is processed by a hard-metal WC-Co with CrN, this will result in a small difference in the need for power in the case of radial and tangential processing.

### 3.5. Cutting Edge Radius Measurement

PVD coatings increases the radius of the cutting edge ϱ. The smallest ϱ value occurs for cutters which are not coated with WC-Co (8 μm). A WC-Co with PVD coating has a bigger radius (WC-Co with CrN—11 μm and WC-Co with TiAlN—10 μm).

### 3.6. Research of Tool Wear

The wear of the rake face of the blades of cutters of WC-Co milling tools and WC-Co milling tools with PVD coatings is presented in Figure 9 (measurement scheme is presented in Figure 2a). The wear of the rake face *Wd* of all of the tested tools changed according to the logarithmical law. The wear occurred suddenly in the working path of 1050–2050 m. It was run-in and later passed into a mode of stable wear. These wear dynamics correspond to the wear indicated in [58]. When the tool exceeded the working path of 9050 m, the rake face of the hard-metal WC-Co cutter was worn by ~ 33 μm, and WC-Co cutters with TiAlN and CrN coatings had wear of up to 22 μm. Greater wear more rapidly changes the angles of the hard-metal WC-Co cutter, the cutting edge is loaded more, it is heated, and its wear becomes more intensive. Thus, the tool's output is potentially smaller. The biggest cause of wear of the rake face of the WC-Co blade is the reduced hardness and bigger COF (Figure 7). Also, the WC-Co surface is susceptible to chemical corrosion from the beginning of the tool's operation. Co, as a binder of WC, is actively affected by the wood extraction substance, tannic acid, so the wear of WC-Co is more active [59]. The CrN coating is more corrosion resistant [50].

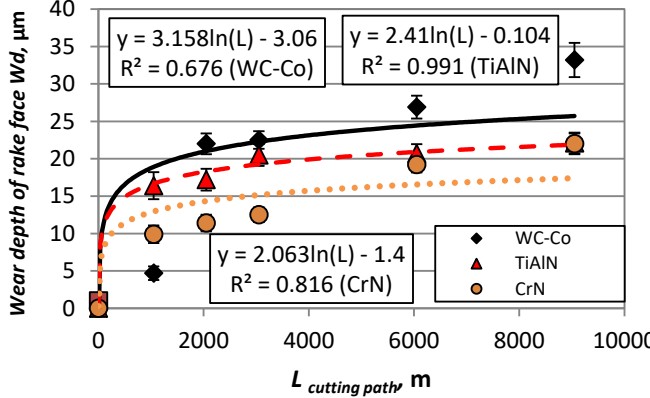

**Figure 9.** Influence of the cutting path *L* on the wear of the rake surface of the hard metal WC-Co cutter with and without PVD coatings when a milling speed of 17 m·s$^{-1}$ is applied on the cutting plane.

The influence of the cutting path *L* (up to 9050 m) on the roughness of the milled surface *Rz* is as follows (Figure 10):

- When the WC-Co tool started working, the roughness was stable and small (4.4 μm). When the working path increased, the roughness of the processed surface increased by up to 5.8 μm; however, it remained smaller than the roughness of surfaces processed by tools with PVD.
- When the cutter coated with CrN started working, the roughness was 5.6 μm. The roughness reached its maximum value (9.2 μm) after the cutter passed 3050 m.

- When the cutter coated with TiAlN started working, the roughness was 6.1 μm. The roughness reached its maximum value (9.5 μm) at 1050 m, and when processing continued, it remained stable and changed in the interval of 5–8.7 μm.

When milling by cutters coated with PVD was started, the processed surface was rougher than in the case of WC-Co. This was attributed to two factors: Firstly, the surfaces of PVD coatings were rougher than the surface of hard metal WC-Co, and secondly, coating with PVD increased the radius of the tool's cutting edge. A negative influence of the static coefficient of friction is possible in the case of TiAlN, because when there is greater friction between the shaving and the cutter's rake face, the shaving is separated from the blank with greater difficulty.

It can be seen that when the WC-Co cutter was used for milling up to 9050 m, the roughness of the processed surface increased. This can be explained by the fact that the wear of the rake surface resulted in an increase in the sharpness angle $\beta$ (reduction of the angle $\gamma$), so sliding the shaving along the rake surface of the tool becomes more difficult, and thus, the roughness increased.

When tools with CrN and TiAlN coatings were used for milling, the roughness of the processed surface changedand had peaks. This could be explained by the fact that damage caused to the rake surface of the PVD coating during processing (Figure 11) made the separation process of the shaving more difficult. According to the SEM of the cutter with CrN (Figure 11b), the damage to the coating was in the form of a step, which aggravated the sliding of the shavings along the rake surface significantly and reduced the processing quality.

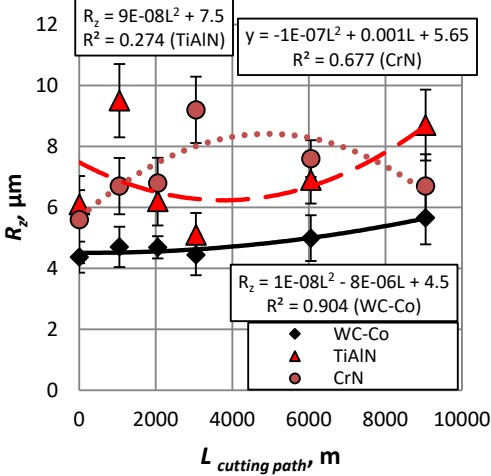

**Figure 10.** Influence of the cutting path *L* on the roughness of the surface $R_z$ (μm) processed with the tool from hard metal WC-Co (with and without PVD coatings)

SEM images of the surface roughness of cutters of WC-Co with and without PVD coatings after the 9050 m path (or after 2.16 million work cycles) are presented in Figure 11.

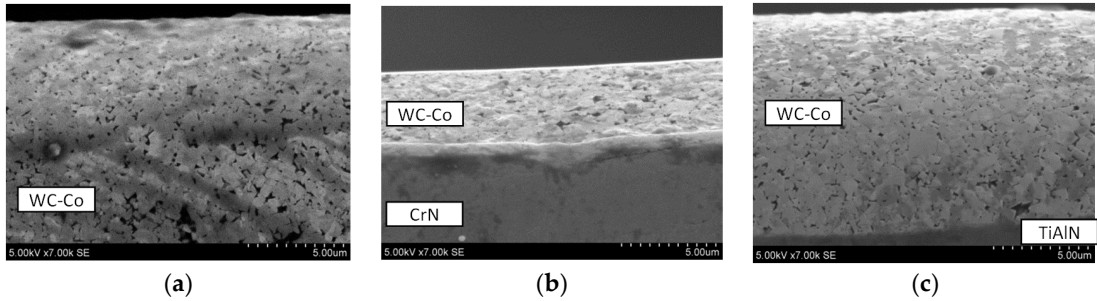

(**a**)          (**b**)          (**c**)

**Figure 11.** SEM images of the rake surface of the cutter of hard metal WC-Co with and without PVD coatings and the cutting edge after the 9050 m path ($2.15 \times 10^6$ work cycles) (×7000): (**a**) WC-Co; (**b**) CrN; (**c**) TiAlN.

The CrN coating was worn on the rake surface by ≈ 4 µm (Figure 11b). The friction of shavings against the rake surface across a width of approximately 55–60 µm resulted in a reduced coating roughness. The cutter's cutting edge was clear (perpendicular). The TiAlN coating was worn by approximately 10–11 µm (Figure 11c), and the surface roughness was reduced under the wear impact across a width of 95 µm (this corresponded to the thickness of the cut shavings). Smooth rounding of the cutting edge was seen. This showed that the CrN coating was more resistant to wear than the TiAlN coating.

The roughness levels of the rake surface of the new cutters and cutters after the 9050 m path are presented in Figure 12.

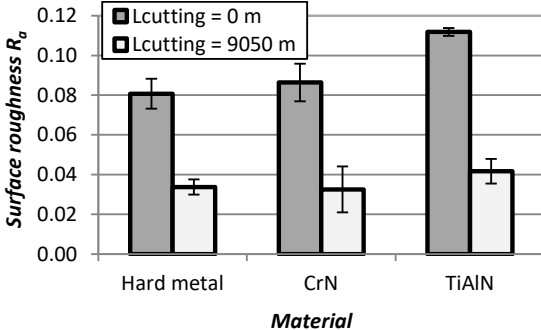

**Figure 12.** Roughness levels of the rake surface (area of cutting edge) of new cutters and cutters after the 9050 m path (with and without PVD coatings).

The roughness levels of the rake surface of new cutters and cutters after the 9050 m path were reduced significantly in the direction perpendicular to the cutting edge (from 2.38 to 2.67 times) (see Figure 12). The surface roughness of the used cutters showed no difference (i.e., in the interval of 0.034–0.042 µm).

## 4. Conclusions

The thickness of the PVD coatings (CrN and TiAlN) was 1.8 to 1.9 µm. The hardness of the hard-metal WC-Co tool was 18 ± 1 GPa; with the CrN coating, it was 19.4 ± 2 GPa, and with TiAlN, it was 30.8 ± 5 GPa. A scratch test helped to determine that the TiAlN coating was damaged by a load of 32.8 N, and the CrN coating was damaged by a load of 44.7 N.

Coating with PVD increased the surface roughness and radius of the cutting edge of the hard-metal WC-Co tool, while the dynamic coefficient of friction was reduced.

A bigger coefficient of friction may cause greater roughness of the processed surface.

In terms of experimental processing in the working path of 9050 m:

- A strip with a width of 4 µm was worn from the cutting edge of the hard-metal WC-Co tool with CrN coating, and a strip of 11 µm in width was worn from the cutter coated by TiAlN;
- The hard-metal WC-Co cutter was worn by 33 µm on the cutting plane, and by 22 µm when CrN and TiAlN coatings were used. All of the tools had active run-in wear in the cutting path of $(1–2) \times 10^3$ m;
- The hard-metal WC-Co cutter processed the oak wood with a roughness of $R_z$ 4.4–5.8 µm, while the cutter with a CrN coating processed the wood with a roughness of 5.6–9.2 µm, and the cutter with a TiAlN coating processed the wood with a roughness of 5.1–9.5 µm.

It is probable that the increased wear of the TiAlN coating was caused by lower resistance to corrosion, worse adhesion parameters, and a bigger static coefficient of friction.

Due to the bigger radius of the cutting edge and the rougher surface of the tool, the cutters with PVD coatings processed the wood more roughly compared to milling using hard-metal WC-Co tools. When the cutters with PVD coatings were used, the roughness was affected by the growing

wear of rake face *Wd* that reduced angle γ. This changed the roughness of the processed surface as the tool became worn.

**Author Contributions:** Conceptualization, D.K. and V.J.; methodology, D.K. and V.J.; software, S.T.; validation, V.J.; investigation, D.K.; resources, D.K.; data curation, V.J.; writing—original draft preparation, D.K.; writing—review and editing, V.J.; visualization, D.K. and V.J.; supervision, V.J.; All authors have read and agreed to the published version of the manuscript.

**Funding:** This research received no external funding.

**Conflicts of Interest:** The authors declare no conflict of interest.

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
