# Peer review of "Research on Tribological Characteristics of Hard Metal WC-Co Tools with TiAlN and CrN PVD Coatings for Processing Solid Oak Wood"

_coatings, doi:10.3390/coatings10070632_

Round 1
Reviewer 1 Report
Some typical questions are as follows:
- There are a lot of mistakes in this manuscript. For example, inconsistent expression, unclear expression, the order in which the graphs appeared (Fig 12 and Fig 13), etc.
- Line 139, author demonstrated TiAlN, but in Fig.4 the diffraction peak marked (AlTi)N2, discussion is not enough.
- In Fig.6(b), the cracks is obvious, but in Line 166, author described “there are no cracks in the coating”.
- It is meaningless to carry out scratch test for uncoated WC-Co substrate. Usually, the scratch test are used to measure the adhesion between coating and substrate.
- The results in Fig 9 are static friction coefficient. How did author test the results?
- Line 215, how can author get the conclusion “the cutting edge of the tool is getting weaker less than in case of uncoated WC-Co tool” ?
- The conclusions are unqualified. It should be simplified further more.
Author Response
We want to thank the reviewers for their valuable comments.
You will find the answers to the comments in the attached document.

Reviewer 2 Report
The subject of paper is scientifically interesting. However in a current form paper contains some flaws and therefore it should be subjected to the major revision. The detailed remarks are as follows:
- In the actual form, the introduction section could be not interesting to readers, because - despite the citation of numeorus of works - it is too short and vague. My recommendation is to avoid blocks of references, e.g. “The cutting tools are under impact of big mechanical and thermal loads, plastic deformations, adhesive and diffusive wear and oxidation [1–10].”, as these do not emphasize the particular aspects from the cited papers. Particularly, when citations are made in reference to specific technical aspects, single/double, e.g. [1, 2] references are encouraged. It is strongly suggested that the references need to make in-depth comments on the content of the cited papers; avoid generic comments. Mention/comment the relevance of the cited paper and especially the research gap associated to it.
- The paper is focused on the machining tests with cemented carbide cutters, however the state-of-the-art concerning this problem matter is very limited in the Introduction section. Thus, the following works should be studied in details and presented in the Introduction section, in order to extend the literature survey:
- 1. Intelligent Optimization of Hard-Turning Parameters Using Evolutionary Algorithms for Smart Manufacturing, Materials 2019, 12, 879; doi:10.3390/ma12060879
- 2. Modeling flatness deviation in face milling considering angular movement of the machine tool system components and tool flank wear. Precision Engineering, 54, 327-337 DOI:10.1016/j.precisioneng.2018.07.001, 2018.
- 3. Tool life and process dynamics in high speed ball end milling of hardened steel. Procedia CIRP 1 (1) (2012) PP. 289 - 294 doi: 10.1016/j.procir.2012.04.052
- 4. The influence of tool wear on the vibrations during ball end milling of hardened steel. Procedia CIRP 14 (2014) 587 – 592
- 5. Effect of the Relative Position of the Face Milling Tool towards the Workpiece on Machined Surface Roughness and Milling Dynamics, Applied Sciences 2019, 9, 842; doi:10.3390/app9050842
- The novelty of the research towards others is not highlighted. Please indicate in the last paragraph of the Introduction, what is the advancement of the current study towards the existing researches in this scope.
- In my opinion, the short section 2 is not needed. Please put the objectives, tasks and motivations in the last paragragh of Introduction.
- P. 3, line 97, the sentence: "The wear of cutter’s rake face wear Wd" should be replaced with: "The wear of cutter’s rake face Wd". Moreover, please check once again carefully the language in the whole manuscript.
- There are some inconsistencies in terminology in paper: during the machining technology the term: "blade" is never being used to describe the working part of the tool. Please replace this term with: "cutting tooth", "working part of the tool", "tool" or the "cutting edge". However, the selection of these terms depends on sentence's context. Please replace the term "cut layer" with "radial depth of cut", "feed per cutter" with "feed per tooth" and "blade radius" with "cutting edge radius".
- the caption of Fig. 2c: " –wood milling scheme" is too general. This is a down milling process. Please justify why this milling kinematics was employed during the machining tests, instead of up milling?
- Table 4: the term "Tool’s rotation frequency n," is inappropriate, the frequency is not expressed in [1/min] but in [Hz]. During the machining processes the term: "tool rotational speed" is usually being employed. Moreover please define (and denote in an appropriate figure) the terms: "Length of contact arch l" and "Shaving’s thickness aΔ".
- According to Authors: "The wear of rake face of the blades of hard alloy WC-Co is the most intensive (it gets worn ~ 33 μm in the milling
208 path of 9050 m). In this case, the wear of rake face Wd is directly proportionate to the working path." I have some doubts concerning this sentence. In my opinion - according to the experimental wear points presented in Fig. 10 - the influence of cutting path on the uncoated WC tool wear is not linear at all! Why Authors have expressed the relation WD=f(L) as a linear function, whereas the curves for the coated tools are expressed by a non-linear functions? In my opinion, all curves should be expressed by the same relations for the comparison purposes. - Please define how the cutting path L - expressed in figure 10 was obtained. Was it expressed in the feed or the main cutting motion (circumferential) direction?
- According to Authors: "When milling by cutters coated with PVD is started, the processed surface is rougher than in case of WC-Co. this could be explained by two reasons: surfaces of PVD coatings are rougher than surface of hard alloy WC-Co, and secondly, coating with PVD increases radius of the tool’s cutting edge. Negative influence of the coefficient of friction is possible in case of TiAlN." In order to confirm this dependency, please provide the measured values of cutting edge radii.
Author Response

(The authors gave the same response as above.)

Reviewer 3 Report
- The abstract section is not a good summary of the present study. Please revise the section carefully. Please begin the abstract section with why the cutting tools were newly studied, and the importance of its application in the industry for which it is intended.
- There are many abbreviations that are not identified in full throughout the article, such as: PVD, CVD, XRD,COF should be clarified.
- Most of the references used are more than 10 years old. There is much more recent literature on the subject to support the study of this article. This situation must be reviewed.
- The study was carried out using PVD deposition however most of the intruduction has to do with the CVD process. In my opinion it should be focused on the process used.
- There is no justification for a topic for the purpose and tasks of the work, it should be presented at the end of the introduction.
- The methodology is not well presented, it has no guiding thread, I cannot understand how many samples were tested and under what conditions, which PVD deposition parameters were used (BIOS, Gas, among others).
- Tables must be formatted according to the format of the journal, the way they are presented makes information difficult to read.
- The strength of this work is undoubtedly the results. They show quite a lot of results, however, it does not cross with the described method. Are the values obtained average values?
- The conclusion is extensive, should be more succinct and clear and should be compared, if possible, with already advanced works.
Author Response

(The authors gave the same response as above.)

Reviewer 4 Report
Manuscript ID: coatings-791537
Research of Tribological Characteristics of Hard Alloy WC-Co Tools with TiAlN and CrN PVD Coatings while Processing Solid Oak-Wood
Interesting job. The subject is of interest and has practical relevance. However, I think the quality of the paper should be improved. For this I would be grateful if you consider the following suggestions:
- Lines 2-3: The authors use the term “hard alloy” for composite material consisting of WC particles bound by a cobalt binder. This term is not common to refer to this type of material. For example, in volume 2 of the ASM Metals Handbook they are called "cemented carbides". In the literature they are also sometimes referred to as "hard metals" such as on the website of the cobalt institute (https://www.cobaltinstitute.org/hard-metal.html). Regardless of these considerations, it is recommended to use the same term throughout the paper. For example, in lines 2-3 it is called “hard alloy”, in line 36 “hard-alloy” and in table 1 “hardmetal”. It is recommended to use the same term in all manuscrispt.
- Lines 21-71: In general, the Introduction is difficult to understand. In some cases, there is a lack of continuity of the sentences. In other cases, it is not understood whether reference is made to PVD coatings or CVD coatings. There are grammatical errors ... It is recommended to make the correction of the English language throughout the paper. Some examples are shown in the following points.
- Line 26: The authors explain that: “The coatings precipitated in PVD mode increase strength of tools”. This sentence is incorrect. PVD coatings do not modify the overall strength of the substrate. I suppose the authors want to indicate that they increase the surface hardness of the tools. Please correct it or explain more clearly what you want to indicate.
- Line 36: “Fine-grained (up to 1 μm) hard-alloy substrate are used.” When are small grain size cemented carbides used? In PVD coatings? In CVD coatings? It is not well understood in the text.
- Line 38: “big hardness at operating temperature, big strength”. Better: "high hardness at operating temperature, high strength"
- Line 41: “The following main characteristics of coatings are stressed:” Better: “The following main characteristics of the coatings are highlighted:”
- Lines 41-64: I do not understand what the authors want to explain. I suppose they mean that coatings have different properties (Lines 42-47), but the most important are the hardness, adhesion and the coating thickness. If this is so, it is recommended to modify the text in order to facilitate its understanding.
- Lines 65-71: I have doubts if it is necessary to include the section "2. Objectives and tasks”. In my opinion, it would be more appropriate to make a critical summary of the state of the art at the end of the section “1. Introduction” and, from this, describe the objectives of the work and the tests / analyzes that have been carried out to achieve these objectives.
- Lines 73-76: The description of the specimens is carried out as if it were a telegram. It would be appreciated if the text could be elaborated a little to make the reading more pleasant.
- Line 80 (Table 1): Units of tensile strength are missing.
- Line 80 (Table 1): What are the indicated units of the coefficient of friction?
- Line 129: It is suggested to include a short description of the characteristics of the coatings shown in figure 3. For example, the columnar structure of the CrN coating explained below. This description would allow a better understanding of the work.
- Lines 133-139: I do not understand what the analysis of the composition of the X-ray diffraction coating contributes to obtain the objectives of the work. Apparently, it only serves to confirm that the company Cemecon Scandinavia has deposited the requested coatings; since the results are not used for a more detailed analysis of the coatings or to discuss the results obtained in the tests. In my opinion, it would be better not to include the section 4.2.
- Lines 145-147: Why does the roughness increase in the coated samples? It would be appreciated if a short explanation is included or some reference from the literature is indicated.
- Lines 169-170: Why does the static coefficient of friction between the CrN coating and oak wood vary with the cutting direction? Curiously, it is the sample in which the roughness is more similar in the direction parallel and perpendicular to the cutting edge. Therefore, it could be thought that it should be the case in which the static friction coefficient is more homogeneous for the different cutting directions. Please discuss the possible reasons.
- Line 211: The authors conclude that “The cause of the biggest wear of rake face of WC-Co blade is smaller hardness and bigger COF”. The hardness of WC-Co and CrN coating is similar. Furthermore, the static coefficient between oak wood and CrN is greater than that between oak wood and WC-Co. However, the wear of the CrN coated tool is less. This, apparently, contradicts the previous conclusion. Please explain the conclusion in more detail.
- Line 213: In the literature, different cases can be found where parabolic behavior of wear evolution is observed. Initially, wear occurs rapidly and progressively decreases the wear rate. In some cases, this is related to a change in the wear mechanism (e. g. adhesive wear vs. mild abrasive wear) and / or the formation of protective layers on the tool surface (e. g. oxide layers). What can be the cause in this case?
- Lines 232: “Negative influence of the coefficient of friction is possible in case of TiAlN.” Why? Please, explain it.
- Line 261: “This shows that CrN coating is more resistant to wear than TiAlN coating” Why? Please, explain it.
- Line 278: “only the belt of 11 μm got worn in the case of cutter coated with TiAlN.” Only? The wear with TiAlN is greater than in the case of the CrN coating. Please write the conclusion more clearly.
Author Response

(The authors gave the same response as above.)

Round 2
Reviewer 1 Report
The manuscript can be accepted in the present form.
Author Response
We wish to thank reviewers for their valuable comments.
We would like to say that the article in English has been checked and corrected by the English native speaker, of the MDPI editor service. We attach english editing certificate.

Reviewer 2 Report
Authors have carefully revised their paper concerning reviewers' remarks. Therefore, its scientific quality has been improved and thus it can be accepted for publication.
Author Response

(The authors gave the same response as above.)

Reviewer 3 Report
I would like to thank the authors, who took into account all the suggestions made. I would also like to congratulate the authors since the article improved significantly when compared to its initial version, it made a very positive leap. The introduction presented is much clearer for the reader and the methodology presents a greater detail, which allows a better understanding of the work described. Regarding the results, they have been improved and show the quality and importance of the work described.
Author Response

(The authors gave the same response as above.)

Reviewer 4 Report
An interesting work. I thank the authors who have taken into consideration the suggestions made by this reviewer. Compared to the initial version, the paper has improved significantly. The introduction is clearer. The methodology used is described in greater detail, which allows a better understanding of the work. The results are also presented more clearly, improving the quality of the paper. However, in the last version of the paper there are some minor errors that should be corrected:
- Line 38: I think that it is not appropriate to name as "more elastic hard metals" those with higher cobalt content. Increasing the cobalt content increases the toughness of the material. It also decreases the Young modulus, but that is not the most important factor in the wear behavior of the material. Therefore, in my opinion it would be more appropriate to name it as “more tough hard metals”.
- Line 89: I think there is an error in the name or units of the property named in the table as "Tensile strength". If the units are MPa m1/2, I think the property is the “Fracture toughness.
- Line 91: Typographic error (Singlelayer)
- Line 196: Typographic error (highlevel)
- Line 283: Typographic error (acrossa)
- Line 309: Typographic error (planeand)
Author Response
Analysis of comments from the reviewers
Research of Tribological Characteristics of Hard Metal WC-Co Tools with TiAlN and CrN PVD Coatings while Processing Solid Oak-Wood
Deividas Kazlauskas, Vytenis Jankauskas, Simona Tučkutė
We wish to thank reviewers for their valuable comments.
Reviewer 4 comments are marked by green color.
We would like to say that the article in English has been checked and corrected by the English native speaker, of the MDPI editor service. We attach english editing certificate.
Reviewer 4 Major comments: |
|||
1 |
Line 38: I think that it is not appropriate to name as "more elastic hard metals" those with higher cobalt content. Increasing the cobalt content increases the toughness of the material. It also decreases the Young modulus, but that is not the most important factor in the wear behavior of the material. Therefore, in my opinion it would be more appropriate to name it as “more tough hard metals” |
We've changed the term to: “more tough hard metals” |
|
2 |
Line 89: I think there is an error in the name or units of the property named in the table as "Tensile strength". If the units are MPa m1/2, I think the property is the “Fracture toughness |
We've changed the term to: Fracture toughness |
|
3 |
Line 91: Typographic error ( Singlelayer) |
We corrected the error to: single layer. |
|
4 |
Line 91: Typographic error ( highlevel) |
We corrected the error to: high level. |
|
5 |
Line 91: Typographic error ( acrossa) |
We corrected the error to: across a ... |
|
6 |
Line 91: Typographic error ( planeand) |
We corrected the error to: plane and ... |
|
